# Disturbance in the protein landscape of cochlear perilymph in an Alzheimer's disease mouse model

**Masatoshi Fukuda[1,2☯], Hiroki Okanishi[3☯], Daisuke Ino[1]\*, Kazuya Ono[1], Satoru Kawamura[1], Eri Wakai[1], Tsuyoshi Miyoshi[2], Takashi Sato[2], Yumi Ohta[2], Takashi Saito[4], Takaomi C. Saido[5], Hidenori Inohara[2], Yoshikatsu Kanai[3,6], Hiroshi Hibino[1,7]\***

1 Department of Pharmacology, Division of Glocal Pharmacology, Graduate School of Medicine, Osaka University, Osaka, Japan, 2 Department of Otorhinolaryngology, Graduate School of Medicine, Osaka University, Osaka, Japan, 3 Department of Bio-System Pharmacology, Graduate School of Medicine, Osaka University, Osaka, Japan, 4 Department of Neurocognitive Science, Institute of Brain Science, Nagoya City University, Nagoya, Aichi, Japan, 5 Lab for Proteolytic Neuroscience, RIKEN Center for Brain Science, Wako, Saitama, Japan, 6 Institute for Open and Transdisciplinary Research Initiatives, Osaka, Japan, 7 AMED-CREST, AMED, Osaka, Japan

☯ These authors contributed equally to this work.
\* dino@pharma2.med.osaka-u.ac.jp (DI); hibino@pharma2.med.osaka-u.ac.jp (HH)

**Data Availability Statement:** The data underlying the supplementary results presented in the study are available from URL: https://repository.jpostdb.org/entry/JPST002363

## Abstract

Hearing loss is a pivotal risk factor for dementia. It has recently emerged that a disruption in the intercommunication between the cochlea and brain is a key process in the initiation and progression of this disease. However, whether the cochlear properties can be influenced by pathological signals associated with dementia remains unclear. In this study, using a mouse model of Alzheimer's disease (AD), we investigated the impacts of the AD-like amyloid β (Aβ) pathology in the brain on the cochlea. Despite little detectable change in the age-related shift of the hearing threshold, we observed quantitative and qualitative alterations in the protein profile in perilymph, an extracellular fluid that fills the path of sound waves in the cochlea. Our findings highlight the potential contribution of Aβ pathology in the brain to the disturbance of cochlear homeostasis.

## Introduction

The cochlea is situated within the intricate folds of the inner ear and plays a key role in hearing. This peripheral organ comprises three tubular structures: the scala vestibuli, tympani, and media (Fig 1). The scala media are immersed in a $K^+$-rich extracellular fluid called endolymph. In contrast, the scala vestibuli and tympani contain perilymph, an extracellular solution with an ionic composition similar to that of general extracellular body fluids, including blood plasma. Perilymph transmits sound waves that propagate through the tubule extending from the scala vestibuli to the scala tympani, thereby initiating the mechano-electrical transduction of auditory information in the sensory epithelium [1,2]. Perilymph interacts with other extra-cellular body fluids, such as cerebrospinal fluid (CSF) and blood. Direct communication

**Funding:** This work was supported by grants from the Ministry of Education, Culture, Sports, Science and Technology (KAKENHI: 21K06421, 23H04167), Inamori Foundation, Brain Science Foundation, Narishige Fund, and Takeda Science Foundation to D.I.; AMED-CREST (23gm1510004) and Moonshot R&D (JPMJMS2024) to H.H. The funders had no role in study design, data collection and analysis, decision to publish, or preparation of the manuscript.

**Competing interests:** The authors have declared that no competing interests exist.

between the perilymph and CSF is facilitated by a narrow bony channel called the cochlear aqueduct, which likely establishes a physical connection between the perilymphatic space of the cochlea and the subarachnoid space of the brain [3]. Although water and small molecules ($< 5$ kDa in molecular weight) can permeate this tiny channel [4–6], the precise nature of the molecules exchanged between these two compartments remains unclear. In addition, perilymph interacts with blood through the blood–labyrinth barrier [7]. This arrangement may allow communication between the cochlea and other organs. Therefore, the inner ear may indirectly communicate with the brain via the peripheral circulatory system, as diverse molecules are exchanged between the blood and the CSF through the blood–brain barrier [8].

Common human brain disorders are associated with hearing impairments [9]. Notably, dementia frequently coexists with hearing loss [10–12]; however, the relationship between these two conditions remains analogous to a "chicken or egg" dilemma. In other words, whether signal transmission from the ear to the brain, from the brain to the ear, or a combination of both underlies the onset and progression of the brain disorder is still unclear. Considering that cerebral atrophy is often detected in patients with age-related hearing loss [13], the prevailing hypothesis is that hearing loss contributes to the advancement of cognitive decline. In rodents, elevating the auditory threshold by exposure to noise or ototoxic drugs induces pathological changes in the hippocampus and cognitive decline [14]. Conversely, whether brain pathologies associated with dementia affect cochlear function remains elusive.

Alzheimer's disease (AD) is the predominant form of dementia and is characterized by progressive memory loss and cognitive decline [15]. A key pathological feature of AD is the accumulation of AD-like amyloid β (Aβ) aggregates in the brain, which exert neurotoxic effects through various mechanisms. These processes include non-specific binding, which leads to cell membrane permeabilization, and specific binding to pattern-recognizing membrane receptors [16–18]. In addition, the effects of brain Aβ pathology on the peripheral tissues, such as the heart, kidney, and liver, have been shown in patients with AD and AD animal models [19–23]. Aβ pathology may also perturbate the homeostasis of the inner ear due to the potential relationship between hearing dysfunction and dementia [24]. Nevertheless, this hypothesis has not been fully proven through experimental approaches.

In this study, we examined how Aβ deposition in the brain affects the cochlear properties using a knock-in mouse model of AD, named $App^{NL-G-F/NL-G-F}$ mice (KI mice). KI mice carry three mutations identified in patients with familial AD: Swedish (NL), Arctic (G), and Beyreuther/Iberian (F) mutations. Introducing these mutations results in the formation of AD-like Aβ aggregations in the brain [25]. Initially, we assessed hearing levels using auditory brainstem response (ABR) measurements and found that age-related shifts in hearing thresholds in KI mice were similar to those in control wild-type (WT) mice. Subsequently, we conducted quantitative and qualitative analyses of perilymph proteins by sampling fluid from live mice at different postnatal ages. The total protein content of the perilymph was approximately 1.5 times higher in KI mice at 6 months of age, when cognitive decline starts to be detectable, than in the WT mice [25]. Moreover, tandem mass tagging (TMT) for quantitative proteomic analysis identified differentially abundant proteins in the perilymph of the two mouse types. Notably, the abundance of proteins potentially related to cellular damage and inflammatory responses was robustly altered in KI mice, suggesting a disturbance in perilymphatic homeostasis. Finally, most altered proteins in the perilymph of KI mice differed from those previously described in the CSF [26]. This observation implies infrequent protein exchange between perilymph and CSF in mice, although conventional belief holds that these two compartments are interconnected with each other. Our findings illuminate a conceivable association between Aβ deposition in the brain and the disturbance of cochlear properties.

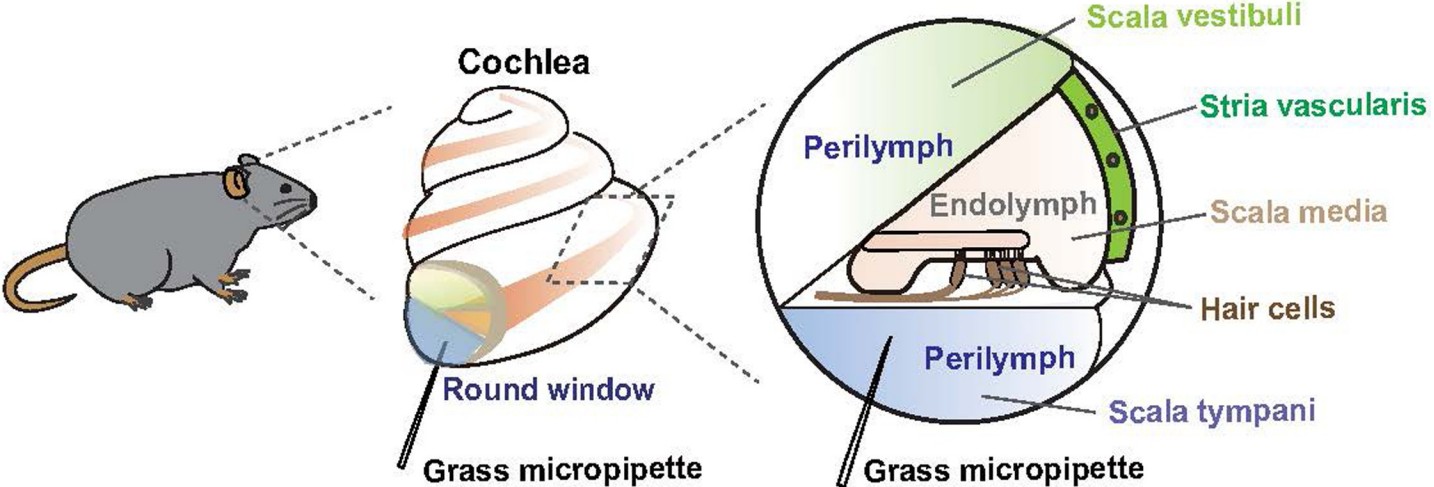

**Fig 1. Structure of mouse cochlea.** An overview of the snail-shaped mouse cochlea (center and left panels) and the internal architecture of the cochlea (right panel) are shown. A glass micropipette was inserted through a round window into the scala tympani to collect the perilymph fluid (center and right panels).

## Materials and methods

### Ethical statement for animal experiments

All animal experiments were approved by the Animal Care and Use Committee of the Osaka University Graduate School of Medicine (Reference Number 03-015-011). The experiments were conducted under the supervision of the committee and per the Guidelines for Animal Experiments of Osaka University and the Japanese Animal Protection and Management Law. All animal handling and reporting complied with the ARRIVE guidelines [27]. All surgeries were performed under anesthesia, and efforts were made to minimize suffering.

### Animals

$App^{NL-G-F/NL-G-F}$ mice (KI), AD knock-in mouse models harboring NL, G, and F mutations in amyloid precursor protein (APP) genes in a homozygous manner [25], were kindly provided by the RIKEN BioResource Research Center (Tsukuba, Japan). The mice were bred with a C57BL/6J background. $App^{wt/wt}$ (WT) and KI mice were generated by mating two heterozygous mutant ($App^{NL-G-F/wt}$) mice. The mice were kept in cages at 23 ± 1.5°C with 45 ± 15% humidity under a regular 12-h dark/light cycle with ad libitum access to food and water. The experimental procedures were conducted during the light phase of the cycle. The animals were randomly assigned to each experimental group without blinding. WT and KI mice of either sex were analyzed at approximately 3, 6, and 12 months of age (n = 8 for each genotype at each age). Anesthesia was induced by intraperitoneal injection of a mixture of medetomidine hydrochloride (0.75 mg/kg; Kyoritsu Seiyaku, Tokyo, Japan), midazolam (4 mg/kg; Sando, Tokyo, Japan), and butorfal tartrate (5 mg/kg; Meiji Seika Pharma, Tokyo, Japan). Anesthetic depth was monitored by assessing the toe pinch responses, corneal reflexes, and respiratory rate as per the guidelines. In cases of insufficient anesthesia, an additional mixture of medetomidine hydrochloride (0.375 mg/kg), midazolam (2 mg/kg), and butorfal tartrate (2.5 mg/kg) was administered. The experiments were only initiated once deep anesthesia was ensured.

## ABR measurements

The anesthetized mice were placed in a prone position in a soundproof booth equipped with acoustic and electrical shielding (Acoustic Systems, Englewood, CO, USA). ABR assessments were conducted using a setup comprising specialized hardware and software (RZ6 Processor and BioSigRZ version 5.7.6 RRID:SCR_014820; Tucker-Davis Technologies, Alachua, FL, USA). Subdermal needle electrodes were inserted at the vertex, under the pinnae of the left ear, and beneath the pinnae of the contralateral ear (ground) of each animal. Cos2-gated 1-ms tone burst stimuli (21/s, alternating polarity) spanning frequencies of 4, 8, 16, or 32 kHz and featuring a 0.1-ms rising, 0.8-ms duration, and 0.1-ms falling phases were delivered to the test ear via a closed-field speaker (MF-1; Tucker-Davis Technologies). Brainstem responses were processed with a band-pass filter (0.3–5 kHz), and 200–500 artifact-free responses were averaged to construct a waveform. For each stimulus frequency, the intensity was initially set to a sound pressure level of 90 dB and gradually reduced in 5 dB increments until the pressure level no longer elicited a discernable waveform. ABR thresholds, representing the minimum stimulus level that induced a detectable response, were determined through visual assessment. The ABR measurements were repeated if the threshold response was unclear. Throughout the measurement, the animals were maintained with spontaneous respiration, and the body temperature was kept near 37°C with a heating pad (Natsume Seisakusho Co Ltd, Tokyo, Japan). Perilymph was collected from mice at 6 months of age, as described in the "Collection of cochlear perilymph from mice" section, whereas other animals were euthanized with an overdose of the anesthetic.

## Collection of cochlear perilymph from mice

After completion of the ABR measurements, each anesthetized mouse was transferred to a head holder, and one side of the cochlea was exposed from the ventrolateral direction [28]. Using a micromanipulator (MP-285; Sutter Instrument Co., Novato, CA, USA), a glass capillary micropipette (World Precision Instruments, Sarasota, FL, USA) was carefully inserted through a round window into the perilymphatic space of the scala tympani in the cochlea (Fig 1). Perilymph spontaneously flowed into the pipette without suction. This method allowed us to obtain 1−2 μL of the fluid from each cochlea in a minimally invasive manner. Upon collecting perilymph on one side, the same procedure was repeated on the opposite side. Perilymph obtained from two mice (i.e., four cochleae) was mixed in a single microtube. Four samples of each genotype were prepared. Before the proteomic analysis, the protein concentration of each sample was determined using a standard Bradford assay by measuring the absorbance at 595 nm (TaKaRa Bio, T9310A, Shiga, Japan). Among the four samples from each WT and KI mouse, the sample with the lowest protein content was excluded from subsequent analyses.

## Proteomic analysis

The samples for liquid chromatograph-tandem mass spectrometry (LC-MS/MS) analysis were prepared as previously described [29,30]. First, the perilymph samples were subjected to heating at 95°C for 5 min in a detergent-containing buffer (100 mM Tris, 12 mM sodium deoxycholate, 12 mM sodium N-lauroylsarcosinate; pH 8.5) to denature the proteins, and debris was subsequently removed through centrifugation at $20,000 \times g$ for 5 min. Subsequently, the proteins in the sample were reduced with 5 mM dithiothreitol (DTT) at 50°C for 30 min and alkylated with 10 mM iodoacetamide at room temperature in the dark for 30 min. The reaction was quenched by adding 5 mM DTT. The proteins were digested with lysyl endopeptidase (Wako) at 37°C for 6 h after a 5-fold dilution with 50 mM ammonium bicarbonate buffer (pH 8.0) and with sequencing-grade trypsin (Promega, Madison, WI, USA) at 37°C overnight.

Finally, the eluted tryptic digest was lyophilized using a centrifugal evaporator CC-105 (TOMY, Tokyo, Japan), followed by dissolution in 10 μL of 10 mM triethylammonium bicarbonate. A TMT 10-plex isobaric label reagent set (Thermo Fisher Scientific, Waltham, MA, USA) in 5 μL of anhydrous acetonitrile was added to label identical TMT-tags to six perilymph samples. After incubation at room temperature for 1 h, to quench the labeling reaction, the samples were mixed with 1 μL of 5% hydroxylamine and incubated for 15 min at room temperature. Then, the labeled samples were mixed in a single tube to quench the labeling reaction.

LC-MS/MS analyses were performed as previously described, with minor modifications [29]. Briefly, the peptides in the samples underwent separation at a flow rate of 300 nL/min on a nano HPLC capillary column (C18, 3 μm, 100 Å pore size, 75 μm i.d., and 120 mm length; Nikkyo Technos, Tokyo, Japan) for 180 min. The eluted peptides were introduced into a Q-Exactive Orbitrap mass spectrometer (Thermo Fisher Scientific) controlled by Xcalibur 4.2 software and analyzed in a scan range of $m/z$ 350–1,500 with a resolution of 70,000 at $m/z$ 200 for MS1. The ions with peak intensities $> 2.0 \times 10^5$ with a charge state from 2+ to 7+ were subjected to MS/MS. In the second MS, the ions were fragmented via higher-energy collision dissociation. Subsequently, the MS and MS/MS spectral data were analyzed by assigning the spectra against the tryptic peptides of *Mus musculus* proteins using Proteome Discoverer 2.3 software (Thermo Fisher Scientific). The search results were filtered through a percolator with a q-value threshold of 0.01. The intensities of the TMT reporter ions with an average reporter S/N > 20 and a co-isolation threshold < 50% were used for quantification. Proteins with two or more unique peptides were used for subsequent analyses. A p-value cut-off and an effect size cut-off were used to define differentially abundant proteins as follows: p-value < 0.05 and fold change either $\geq 1.5$ (for upregulated) or $\leq 0.67$ (for downregulated). This threshold is widely used to determine whether a protein is upregulated or downregulated [31–33]. Protein localization was defined based on the UniProt annotation. Comparable data for CSF proteomic analysis were extracted from a previous study to compare differentially abundant proteins in the perilymph of KI mice with those in the CSF [26].

## Statistical analyses

All data are expressed as the mean ± standard error of the mean (SEM). Two-tailed Student's t-tests were used for comparisons between the two groups. One-way analysis of variance (ANOVA), followed by Bonferroni's *post-hoc* test, was conducted when comparing more than two groups. Data distribution was assumed to be normal, and the variance was similar between the groups that were statistically compared. No statistical methods were used to predetermine the sample sizes; however, our sample sizes were similar to those generally used in the field [34–36]. Differences between groups with p < 0.05 were considered statistically significant. Microsoft Excel 2019 (Microsoft, Seattle, WA, USA) and Adobe Illustrator 2023 (Adobe, San Jose, CA, USA) were used to create graphs. Microsoft Excel 2019 was used to calculate statistical parameters.

## Results

### Effect of brain Aβ pathology on hearing threshold

We compared the age-related changes in KI mice with those in WT mice to examine whether AD-like Aβ pathology in the brain impacts hearing. As age-related hearing decline starts at around 3 months and gradually progresses after that in this genotype, we measured the ABR thresholds of either genotype at 3, 6, and 12 months of age [37]. Age-related hearing impairment was evident across all measured frequencies (4, 8, 16, and 32 kHz) in KI mice; this

profile was similar to that observed in WT mice (Fig 2). Moreover, in all three age groups, the auditory thresholds at any frequency in the KI mice were comparable to those in the WT mice. These results suggest that the accumulation of pathological Aβ aggregates in the brain of KI mice is unlikely to accelerate hearing loss.

### Relevance of brain Aβ pathology to the protein profile of cochlear perilymph

Next, we elucidated whether any other elements in the cochlea would be affected by AD-like Aβ pathology in KI mice. Because perilymph potentially communicates with the CSF and blood plasma [4,5,38], pathological signals are possibly transmitted from the brain to the cochlea directly or via peripheral circulation in KI mice. To test this hypothesis, perilymph was collected from the cochleae of live mice (Fig 1). First, we measured the protein content of the samples at three, six, and 12 months of age (Fig 3). In WT mice, the concentration gradually increased with age, whereas in KI mice, the concentration peaked at 6 months of age. Notably, at this time point, the protein content in KI mice was approximately 1.5 times greater than that in WT mice (n = 4 samples for each genotype, 1.1 ± 0.14 mg/mL for KI, 0.70 ± 0.05 mg/mL for WT mice, respectively, p = 0.045). At 3 and 12 months, little difference was detected between WT and KI mice (n = 4 samples for each genotype at each age, 3 months: 0.63 ± 0.18 mg/mL for KI, 0.66 ± 0.13 mg/mL for WT, p = 0.87; 12 months: 1.1 ± 0.15 mg/mL for KI, 1.1 ± 0.2 mg/mL for WT, p = 0.77). Consequently, the most substantial disparity between KI and WT mice likely occurred at 6 months of age.

Accordingly, we focused on 6 months of age and compared perilymph between KI and WT mice using TMT-based proteomic analysis (Fig 4A). We identified 18 robustly upregulated proteins and 21 markedly downregulated proteins in KI mice (Fig 4B and 4C). To characterize these proteins, we categorized their localization into five classes: secreted, extracellular, membrane, cytoplasm, and mitochondrial matrix, based on annotations in UniProt (Fig 4D). Approximately 70% of the upregulated proteins (13 of 18 molecules) were associated with cellular components (extracellular, membrane, cytoplasm, or mitochondrial matrix). This observation suggests that the increase in protein levels in KI mice was primarily attributable to the leakage of cellular components into the perilymph. The remaining 30% (5 of 18 molecules)

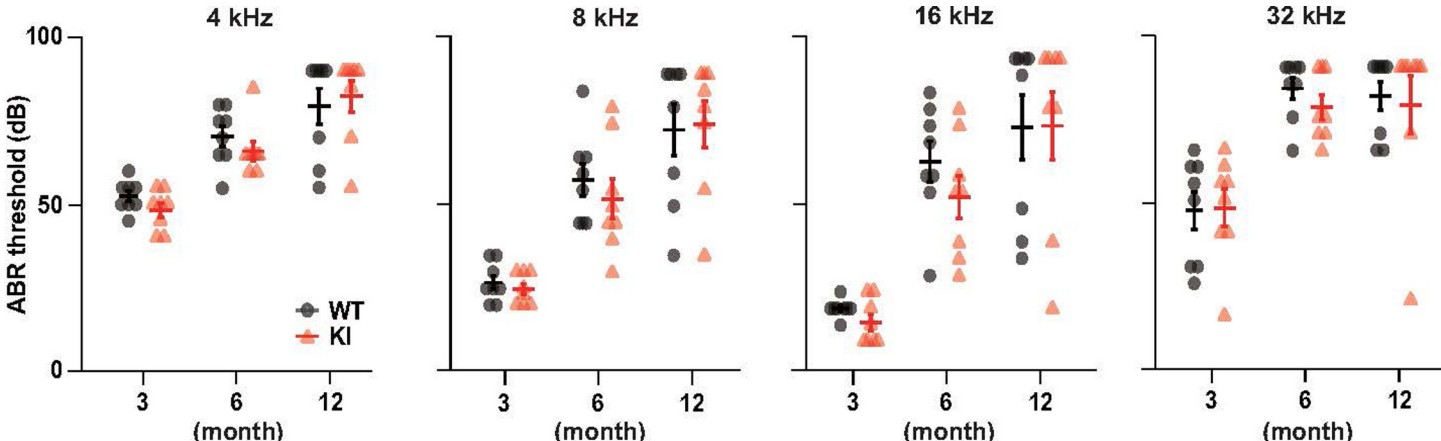

**Fig 2. Auditory brainstem response (ABR) thresholds of wild-type (WT) and $App^{NL-G-F/NL-G-F}$ (KI) mice.** WT and KI mice at 3, 6, and 12 months of age (n = 8 for each group) were subjected to ABR measurements, and auditory thresholds at 4, 8, 16, and 32 kHz were determined. All the data recorded at each frequency are presented in a panel with mean ± SEM. Statistics: one-way ANOVA ($F_{5,42}$ = 16, p = 1.1 ×$10^{-8}$ for 4 kHz; F = 16, p = 6.7 ×$10^{-9}$ for 8 kHz; F = 15, p = 2.2 ×$10^{-8}$ for 16 kHz; F = 9.9, p = 2.8 ×$10^{-6}$ for 32 kHz) with Bonferroni *post-hoc* test (p = 1 for all comparisons between WT and KI with the same age and the frequency).

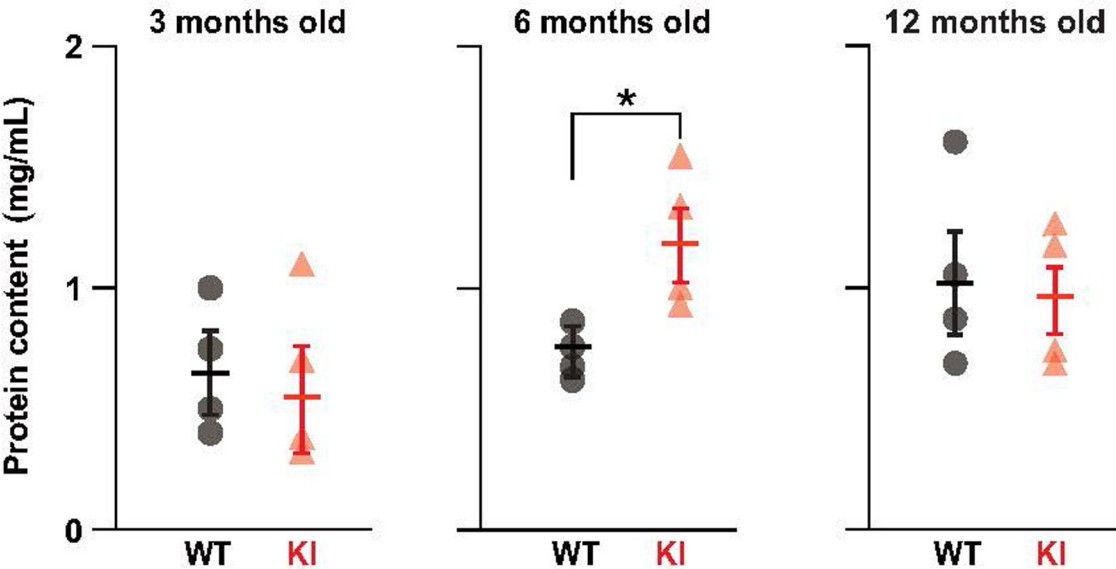

**Fig 3. Protein content of the perilymph obtained from WT and KI mice.** The concentrations of the perilymph from WT and KI mice at 3, 6, or 12 months of age (n = 4 samples for each group) were measured; all the data are plotted in each panel with mean ± SEM. Statistics: two-tailed Student's t-test (p = 0.65, 3 months; p = 0.045, 6 months; p = 0.78, 12 months). *p < 0.05.

were secreted proteins, including components involved in the innate immunity complement system, such as complement proteins C8 (CO8B) and C9 (CO9). Activating the immune system involves elevating complement proteins [39,40]. Therefore, in KI mice, inflammatory responses likely extend into the cochlea at 6 months when the robust AD-like Aβ pathology is present in the brain [25]. In addition, a considerable number of perilymphatic proteins were downregulated in KI mice (Fig 4B and 4C). In this context, roughly half (11 of 21 proteins) were associated with cellular components. The rest (10 of 21 proteins) were classified as secreted proteins (Fig 4D). Notably, downregulated proteins included binding proteins for insulin-like growth factor 1 (IGF-1), such as IBP4, IBP5, and IBP6 [41,42] (Fig 4C). Since IGF-1 likely protects the cochlea from chronic inflammation and cell death [43,44] in KI mice, the decrease in signaling molecules related to IGF-1 may mirror the disturbance in cochlear homeostasis.

## Differential impact of brain Aβ pathology on perilymph and CSF

We then compared our perilymphatic proteomic data with the CSF data previously described to gain insight into how the protein profile in the cochlea in KI mice was disturbed by the AD-like Aβ pathology in the brain [26]. Among the proteins that were quantitatively altered in the CSF of KI mice compared to WT mice (14 upregulated and 22 downregulated proteins), only a marginal subset (one upregulated and two downregulated proteins) overlapped with the catalog of our perilymph analysis (Fig 5A and 5B). Therefore, changes in the CSF profile are unlikely to contribute directly to changes in the perilymphatic profiles of KI mice. Nevertheless, similar to our data (Fig 4D), more than half of the upregulated and downregulated proteins in the CSF were cellular components (~60%: 8 of the 14 upregulated proteins and 13 of the 22 downregulated proteins, respectively) (Fig 5C). The contrasting results in the perilymph and CSF indicate that the sources of the differentially expressed proteins in the extracellular fluid may be distinct between the cochlea and brain in KI mice.

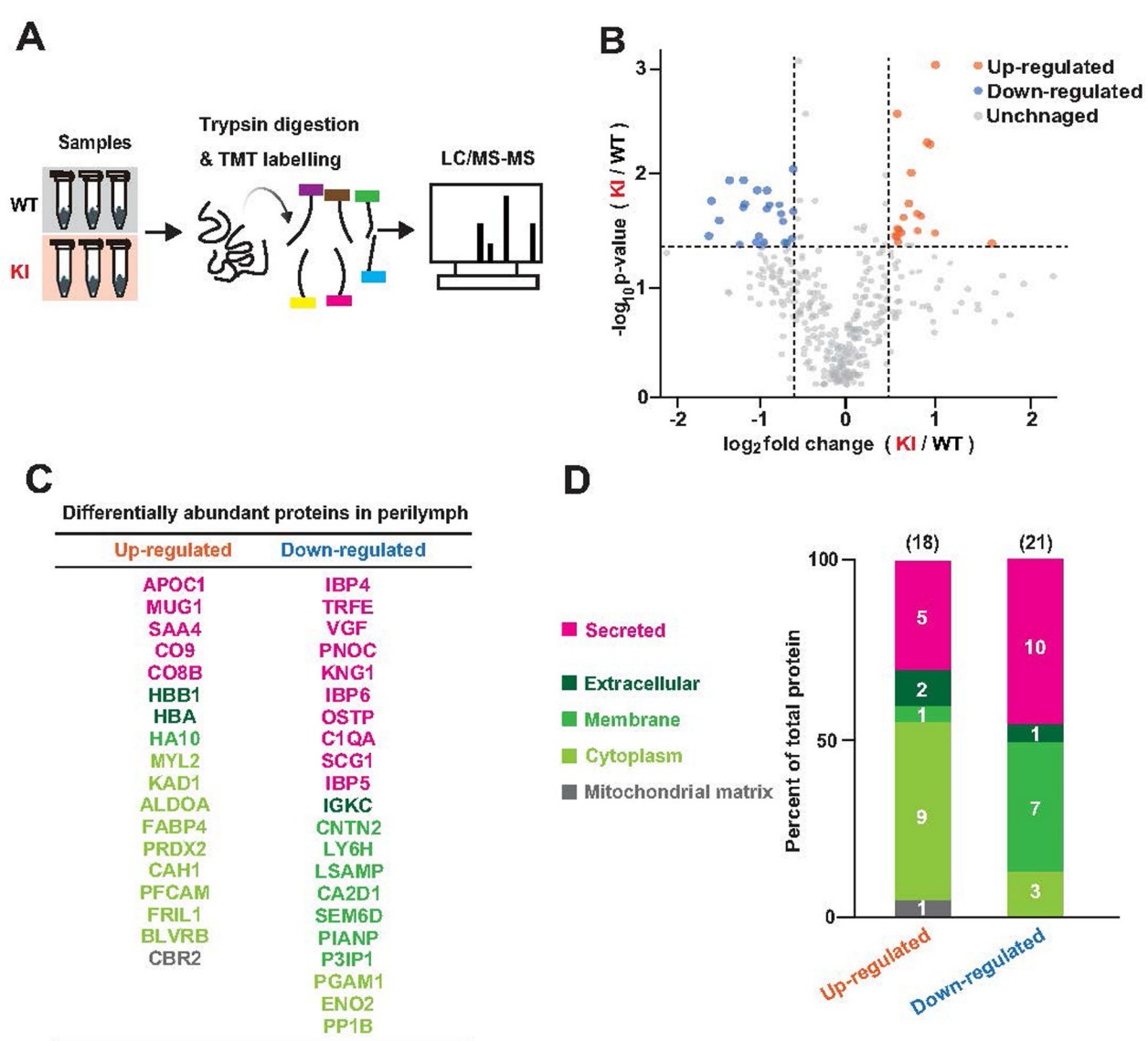

**Fig 4. Comparison of the landscape of the perilymphatic proteins between WT and KI mice.** (A) Schematic illustration of tandem mass tagging (TMT) proteomic analysis of the perilymph samples. (B) Volcano plots displaying the differentially abundant proteins in the perilymph of KI and WT mice. The horizontal dotted line represents the cut-off p-value (0.05), whereas the vertical dotted lines show cut-off thresholds for downregulated (0.67) and upregulated proteins (1.5). (C and D) List and characterization of the quantitively changed proteins in KI mice compared to that in WT mice. The names of the up- or downregulated proteins are shown in panel C. In panel D, these proteins were classified in terms of the localization to the five different categories according to the annotation in UniProt.

## Discussion

In this study, we aimed to address whether and how AD-like Aβ pathology in the brain can influence the cochlea by analyzing KI mice, an AD model that harbors multiple disease-related mutations in *App* and replicates a key AD-like pathology (i.e., Aβ deposition) in the brain [45–47]. Age-related changes in hearing thresholds in KI mice were similar to those observed in WT mice. We also explored perilymph and identified discernible alterations in the protein landscape in KI mice at 6 months of age compared to their counterparts in WT mice.

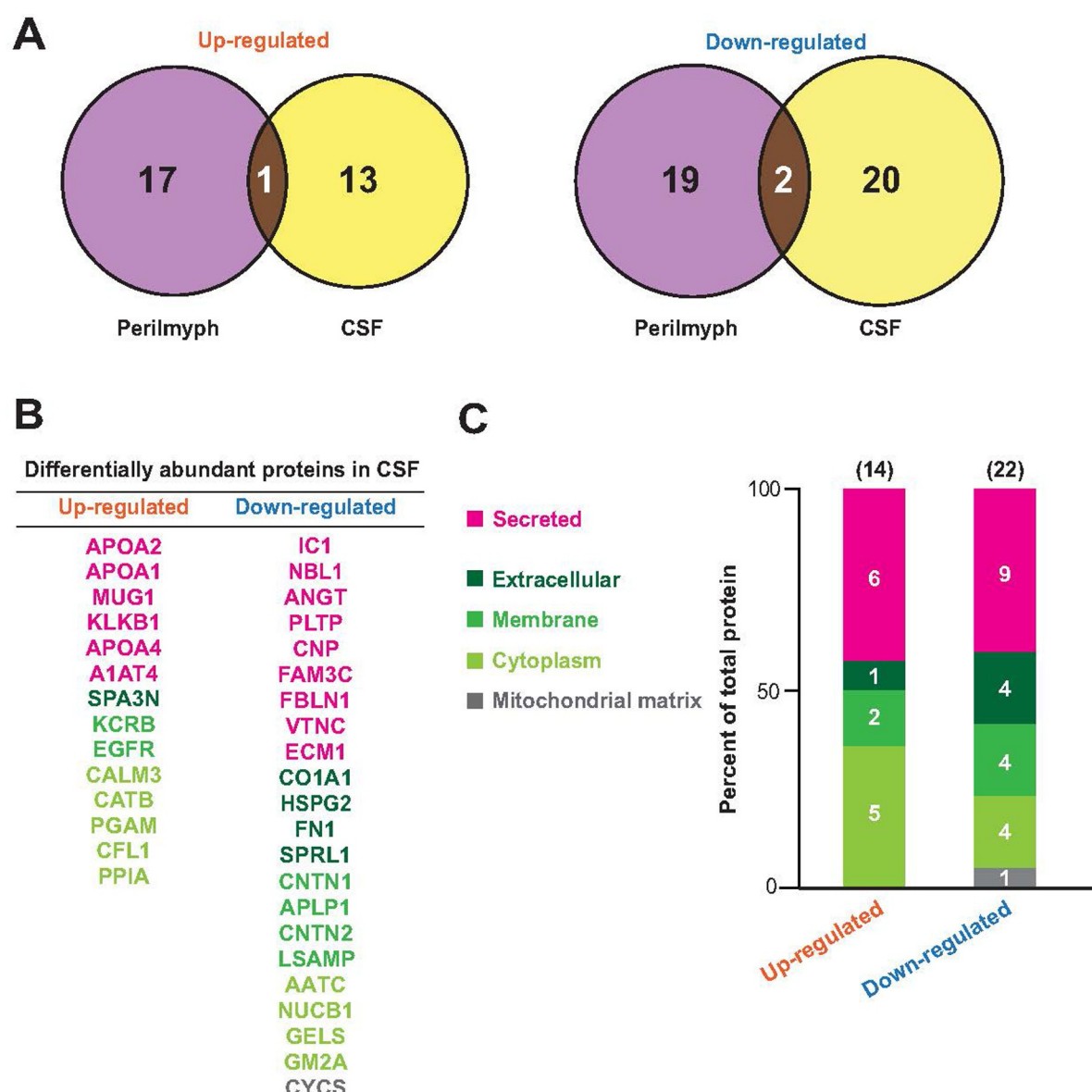

**Fig 5. Protein profiles of the perilymph and cerebrospinal fluid (CSF) in KI mice.** (A) Venn diagrams showing the numbers of up- or downregulated proteins in perilymph or CSF (left: upregulated, right: downregulated). (B and C) List and characterization of the quantitively changed proteins in the CSF of KI mice compared with that in WT mice. The names of up- or downregulated proteins are displayed in panel B. In panel C, the proteins are classified in terms of localization into five different categories according to the annotation in UniProt. The data are derived from the work of Jiang et al. [26].

Epidemiological studies in humans have demonstrated a potential link between dementia and hearing loss [48–50]. In a different AD model, termed 3×Tg-AD mice, hearing loss induced by acoustic trauma causes inflammatory reactions and synaptic dysfunction in the hippocampus and promotes cognitive decline [51]. In contrast, the similarity in the ABR thresholds between KI and WT mice at any postnatal age suggests that Aβ-deposition in the brain negligibly contributes to age-related hearing loss progression. Nonetheless, in APP/PS1 mice, an alternative AD model that overexpresses chimeric mouse/human APP and mutant human presenilin 1 (PS1) under the control of mouse prion protein (PrP) promoter elements,

the hearing threshold is elevated by approximately 20 dB sound pressure level compared to the threshold in WT mice as early as 3 months of age [11]. Because this phenotype precedes a significant accumulation of Aβ plaques in the brain (6 months old), the hearing impairment is unlikely to be attributed to brain pathology [52]. Given that the induction of transgenes by the PrP promoter is not exclusive to brain cells [53], expression of APP, PS1, or both outside the brain may affect the auditory tract.

We observed quantitative and qualitative changes in perilymphatic proteins in KI mice. However, how this disturbance is induced is unclear. Approximately 70% of the upregulated proteins belonged to the cellular components category encompassing extracellular, membrane, cytoplasm, and mitochondrial matrix. This result suggested that the elevated protein fractions in the perilymph of KI mice are likely released from damaged cells in the cochlea, brain, or peripheral tissues. First, because the age-related progression of hearing loss was not accelerated in KI mice, damage to the cochlea was likely minimal. Second, the upregulated cellular components in the perilymph of KI mice encompassed a few proteins specific to peripheral tissues, such as myosin regulatory light chain 2, fatty acid-binding protein 4, and carbonic anhydrase 1 (CAH1) [54] (Fig 4C and 4D); these proteins should be transported from the peripheral tissues to the perilymph via peripheral circulation. Finally, in KI mice, only a few up- or downregulated proteins overlapped between the perilymph and CSF (Fig 5), likely excluding the direct transport of proteins between these fluids. Therefore, we can hypothesize that the changes in perilymphatic protein composition in KI mice are mediated primarily by transferring cellular proteins from blood plasma to the cochlea via the blood–labyrinth barrier [55]. Then, what is the pathological significance of disturbances in the perilymphatic protein profile of KI mice? Notably, the levels of complement proteins, which are potential inflammatory factors, were elevated. This observation suggests that at 6 months of age, the cochlea in KI mice may be engaged in inflammatory responses to some extent. Acoustic trauma induces inflammation in the perilymph, and this pathological state is likely involved in hearing impairment [56]. Similarly, chronic inflammation in the cochlea of KI mice may result in the mice being more susceptible to hearing loss upon being subjected to external stresses, although the animals exhibited apparently normal hearing thresholds (Fig 2). To prove this idea, further studies are needed to clarify the relationship between perilymphatic protein repertoire and cochlear function.

Although perilymph and CSF are physically interconnected via the cochlear aqueduct [4–6], whether the protein profiles in the two compartments are similar is uncertain. Therefore, we compared the results from our perilymph proteomic analysis with the corresponding data in the CSF [26]. Only a few proteins were shared between perilymph and CSF, suggesting minimal protein exchange between the two fluids in mice despite the cochlear aqueduct. Furthermore, in WT mice, the perilymph protein content is approximately three times higher than that of the CSF [57]. These observations support the hypothesis that the perilymph has a protein environment that is relatively independent of the CSF, at least in mice. Nevertheless, trypan blue, a water-soluble dye with a molecular weight of < 5 kDa, reaches the brain when injected into the inner ear [3]. Therefore, small molecules such as sugars, amino acids, and nucleic acids may permeate the cochlear aqueduct in mice, affecting cochlear cells, and modulating the protein profile of the fluid. Notably, the aqueduct is larger in humans (~1 mm in diameter) than in mice [58,59]. Therefore, in humans, the cochlear aqueduct may serve as a passage for macromolecules, including proteins.

This study provides informative data on the cochlear environment in the AD mouse model, but has a few limitations. First, the perilymph samples collected from mice of both sexes were mixed and analyzed (Fig 4); this approach may involve a confounding factor. Nevertheless, in each series of the experiments, the genetic background of the animals we used and their age were unified (i.e., C57BL/6J and the ages of 3, 6, and 12 months old) to minimize effects other

than AD-like Aβ pathology. Second, because the volume of perilymph in a cochlea is small, in the sampling of this fluid we could not completely rule out the possibility of contamination by cellular components and/or other extracellular solutions. The third issue is the relatively small sample size used in the study (n = 3; see Fig 4). Nonetheless, previous studies reporting the results of TMT analysis have conducted similar experiments with the same number of biological replicates as that used in our work to examine statistically significant differences among multiple subject groups [29,32]. This possible limitation is also related to the technical difficulty involved in our procedure and ethical concern for animal experiments. In a mouse cochlea, the volume of perilymph is extremely small, approximately 5 µL [60]. Therefore, collection of this fluid with minimal contamination of other fluids and cells requires a high level of skill. In other words, increasing the sample size must be accompanied by a significant increase in the number of animals we test. In this context, we need to pay attention to such ethical concerns that, in general, the number of animals used in any study must be minimized as much as possible. Taken together, we concluded that the sample size described in this study is acceptable. Finally, although the results detected in the perilymph of KI mice by LC-MS/MS should be validated by other approaches such as western blot and/or ELISA analyses, we did not conduct these assays. This strategy is attributed to our standpoint of minimization of the number of the tested mice as mentioned above.

In conclusion, our findings reveal the potential impact of AD-like pathology in the brain on the homeostasis of perilymphatic proteins in the cochlea. Taking this into consideration, it is of interest to identify the mechanisms underlying the up- and downregulation of the perilymphatic proteins in KI mice and the pathophysiological roles that these proteins play in cochlear activity and brain function. In particular, the latter issue must be crucial from a clinical standpoint. By clarifying the relationship between hearing loss and AD in mice, we were able to apply our findings to the establishment of a diagnostic biomarker for AD-related hearing impairment in the future, although collecting perilymph from human cochlea is challenging and invasive under the existing clinical situation. Considering the relationship between different neurodegenerative diseases and hearing impairment [48,61–64], disturbances in the perilymph profile may be common across patients with several brain disorders. A possible target is Parkinson's disease; this disease is related to a-synuclein deposits in the brain, reminiscent of Aβ deposits in the brain of patients with AD. Indeed, in this study a-synuclein was found in perilymph, although little change was detectable between the samples taken from WT and KI mice. Further studies are required to uncover the pathophysiological significance of brain-to-perilymph remote signaling.

## Supporting information

**S1 File.**
(DOCX)

## Acknowledgments

We thank all members of the Hibino laboratory for their helpful discussions, Ms. Y. Mizuno for her technical assistance, and Drs. E. Oiki and K. Niwa at the Center for Medical Research and Education, Graduate School of Medicine, Osaka University, for their support in measuring protein content.

## Author Contributions

**Conceptualization:** Daisuke Ino.

**Investigation:** Tsuyoshi Miyoshi.

**Project administration:** Takashi Saito, Takaomi C. Saido.

**Supervision:** Hiroki Okanishi, Daisuke Ino, Kazuya Ono, Takashi Sato, Yumi Ohta, Hidenori Inohara, Yoshikatsu Kanai, Hiroshi Hibino.

**Visualization:** Satoru Kawamura, Eri Wakai.

**Writing – original draft:** Masatoshi Fukuda, Eri Wakai.

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
