## [Decision Letter · Decision Letter 0]

4 Mar 2024

PONE-D-24-02136Disturbance in the protein landscape of cochlear perilymph in an Alzheimer's disease mouse modelPLOS ONE

Dear Dr. Hibino,

Thank you for submitting your manuscript to PLOS ONE. After careful consideration, we feel that it has merit but does not fully meet PLOS ONE’s publication criteria as it currently stands. Therefore, we invite you to submit a revised version of the manuscript that addresses the points raised during the review process.

We look forward to receiving your revised manuscript.

Kind regards,

Abdelwahab Omri, Pharm B, Ph.D, Laurentian University

Academic Editor

PLOS ONE

Additional Editor Comments:

The manuscript presents valuable insights into the impact of AD-like pathology on the cochlear perilymph protein profile. While the study has several strengths, including comprehensive methodology and a clear presentation of results, there are areas for improvement, such as increasing the sample size and providing mechanistic insights. Addressing these limitations could enhance the significance and impact of the study.

**Strengths:**

**Comprehensive Methodology:** The manuscript provides a detailed description of the experimental procedures, including the collection of perilymph, protein analysis using liquid chromatograph-tandem mass spectrometry (LC-MS/MS), and statistical analyses. This thorough methodology enhances the credibility of the study's findings.

**Clear Presentation of Results:** The results are clearly presented, with figures and tables aiding in the interpretation of data. The use of TMT-based proteomic analysis allows for a comprehensive evaluation of protein changes in the cochlear perilymph.

**Insightful Discussion:** The discussion section effectively interprets the results in the context of existing literature, highlighting the potential implications of the findings for understanding the impact of Alzheimer's disease (AD)-like pathology on cochlear function. The discussion also raises important questions for future research.

**Area of improvement:**

**Limited Sample Size:** The manuscript mentions using four samples for each genotype at each age group, which may be considered relatively small for proteomic analysis. Increasing the sample size could enhance the robustness of the findings and improve statistical power.

**Lack of Mechanistic Insight:** While the study identifies changes in the protein profile of cochlear perilymph associated with AD-like pathology, it does not delve deeply into the underlying mechanisms driving these alterations. Including mechanistic insights could strengthen the significance of the findings.

**Validation of Findings:** To strengthen the validity of the observed protein changes, additional validation experiments such as immunohistochemistry or functional assays could be performed.

**Inclusion of Longitudinal Data:** Including longitudinal data on protein changes in the perilymph over time could provide insights into the progression of cochlear dysfunction in AD models.

**Clarification of Clinical Relevance:** The manuscript could benefit from further discussion on the potential clinical relevance of the findings for understanding the relationship between AD and hearing impairment in humans.

**Questions:**

1. Can you provide further details on the rationale for selecting specific proteins for analysis in the cochlear perilymph, and how were these proteins chosen?

2. Could you elaborate on the potential implications of the observed changes in protein profiles in the context of cochlear function and AD pathology?

3. How do you address the potential limitations associated with the relatively small sample size used in the study?

4. Can you discuss any potential confounding factors that may have influenced the observed protein changes in the cochlear perilymph?

5. Can you provide insights into the mechanisms underlying the observed alterations in protein composition in the cochlear perilymph in response to AD-like pathology?

6. How do you envision translating these findings into potential therapeutic strategies or diagnostic biomarkers for AD-related hearing impairment?

7. Have you considered conducting additional experiments to validate the observed protein changes using alternative methods or approaches?

8. Could you discuss the relevance of the observed protein changes in the cochlear perilymph to other neurodegenerative diseases beyond AD?

9. What are the next steps in your research agenda to further elucidate the relationship between AD pathology and cochlear dysfunction, based on the findings of this study?

Reviewers' comments:

Reviewer's Responses to Questions

**Comments to the Author**

1. Is the manuscript technically sound, and do the data support the conclusions?

Reviewer #1: Yes

2. Has the statistical analysis been performed appropriately and rigorously? 

Reviewer #1: Yes

3. Have the authors made all data underlying the findings in their manuscript fully available?

Reviewer #1: Yes

4. Is the manuscript presented in an intelligible fashion and written in standard English?

Reviewer #1: Yes

5. Review Comments to the Author

Reviewer #1: The author in this research study is trying to establish correlation between Alzheimer's disease and hearing impairment by inducing disease in the animal model, The experimental work conducted is satisfactory. However further studies can be conducted to human beings to substantiate the finding s and confirm the proof of concept

6. PLOS authors have the option to publish the peer review history of their article (what does this mean?). If published, this will include your full peer review and any attached files.

Reviewer #1: **Yes: **Sujata Pralhad Sawarkar

---

## [Author Response · Author response to Decision Letter 0]

14 Apr 2024

Apr 3rd, 2024

Dear Dr. Omri

My co-authors and I are pleased to submit our revised manuscript entitled ‘Disturbance in the protein landscape of cochlear perilymph in an Alzheimer's disease mouse model’ (PONE-D-24-02136), for your consideration for publication in PLoS ONE. The comments and concerns raised by both you and the reviewer were very constructive and helpful. Based on your suggestions, we have carefully revised the text and figures. Our point-by-point responses to your and the reviewer’s comments are included below.

We thank you very much for your kind remarks and hope that you and the reviewers find that our revised version is improved and now suitable for publication in PLoS ONE.

Yours sincerely,

Hiroshi Hibino

Professor & Chairman

Division of Glocal Pharmacology

Department of Pharmacology

Graduate School of Medicine

Osaka University

2-2 Yamadaoka, Suita, 

Osaka 565-0871, Japan

E-mail: hibino@pharma2.med.osaka-u.ac.jp

Tel.: +81 6-6879-3512 

Fax: +81 6-6879-3519

 

Responses to Editor’s Comments

1. Please ensure that your manuscript meets PLOS ONE’s style requirements, including those for file naming.

In this revision, we arranged the manuscript according to the PLoS ONE style. We set the 1st heading’s font size to 18pt, the 2nd heading's font size to 16pt, and inserted an indent at the beginning of each paragraph. We also submitted specific figures in EPS format, dividing each figure separately.

We checked the reference style used in the manuscript and verified the availability of all the references. We changed the format to Vancouver style, deleted the reference numbered 41, and added new references 31, 32, 33, 57, 61, 62, 63. Citation numbers in the manuscript have also been changed appropriately.

Responses to Additional Editor’s Comments

Area of improvement:

Limited Sample Size: The manuscript mentions using four samples for each genotype at each age group, which may be considered relatively small for proteomic analysis. Increasing the sample size could enhance the robustness of the findings and improve statistical power.

 We agree with your concerns regarding the sample size used in our experiments. We concluded that the sample size was reasonable and in accordance with the number of animals used in a few previous studies. In addition, the technical difficulty involved in our procedure and ethical concerns for experimental animals were critical factors for the small sample size. For details, please see our response to your Question 3 described below.

Lack of Mechanistic Insight: While the study identifies changes in the protein profile of cochlear perilymph associated with AD-like pathology, it does not delve deeply into the underlying mechanisms driving these alterations. Including mechanistic insights could strengthen the significance of the findings.

 I appreciate your criticism. We think that the protein profile in perilymph would be altered by means of a route via blood or cerebrospinal fluid as described in our response to Question 5.

Validation of Findings: To strengthen the validity of the observed protein changes, additional validation experiments such as immunohistochemistry or functional assays could be performed.

 Thank you for this valuable comment. As mentioned in our responses to Questions 3 and 7, in this study we made every effort to minimize the number of mice used for our experiments due to technical difficulties in obtaining cochlear fluid and ethical concerns. Therefore, we did not carry out other additional experiments. We will take your suggestions into consideration in our future studies.

Inclusion of Longitudinal Data: Including longitudinal data on protein changes in the perilymph over time could provide insights into the progression of cochlear dysfunction in AD models.

 We completely agree with your comment. Because of the small volume of perilymph in a cochlea and the technical difficulty involved in sampling this fluid, such additional experiments will require numerous animals. From a standpoint of ethical concern, we did not carry out further experiments because we intended to minimize the number of animals used in this study. For more details regarding this issue, please see our response to Question 3 as well.

Clarification of Clinical Relevance: The manuscript could benefit from further discussion on the potential clinical relevance of the findings for understanding the relationship between AD and hearing impairment in humans.

 We appreciate your constructive comment. We would also like to apply our result to elucidation of the relationship between AD and hearing impairment in humans. As written in our response to Question 6, in the revised version of the manuscript I have added a few sentences discussing the possibility of the application of our methods and results to humans.

Response to Questions:

1. Can you provide further details on the rationale for selecting specific proteins for analysis in the cochlear perilymph, and how were these proteins chosen?

We appreciate your valuable comment. Several specific proteins are reported to be increased in the cerebrospinal fluid and blood in patients with AD or AD mouse models [1, 2]. Therefore, in this study, we focused on the cochlea of the inner ear in an AD mouse model and examined the change of the protein landscape in an extracellular fluid, perilymph. The criteria that we used for the selection of differentially abundant proteins are a p-value cut-off of < 0.05 and an effect size cut-off as follows: fold change of either ≥ 1.5 (for up-regulated) or ≤ 0.67 (for down-regulated) (see page 9 lines 194 − 196 in the Methods section). In order to demonstrate the widespread use of these standards, citations to relevant literature have been added [3-5] with an explanatory sentence (see page 9 lines 196 − 198 in the Methods section).

2. Could you elaborate on the potential implications of the observed changes in protein profiles in the context of cochlear function and AD pathology?

 We thank you for your constructive comment. Because the proteins elevated in the perilymph of KI mice at 6 months old contained complement proteins C8 and C9, AD-like Aβ pathology in the brain could be associated with the induction of inflammation in the cochlea. Acoustic trauma induces inflammation in the perilymph and this pathological state is likely involved in hearing impairment [6]. Similarly, chronic inflammation in the cochlea of KI mice may possibly cause the mice to be more susceptible to hearing loss upon being subjected to external stresses, although they apparently exhibit normal hearing thresholds. In the revised text, we added this discussion to page 15 lines 316 − 320 in the Discussion section.

3. How do you address the potential limitations associated with the relatively small sample size used in the study?

We agree with your concern that the sample size in our study was relatively small. The number of samples was determined in accordance with the following two issues. First, there are a number of papers reporting the results of TMT analysis conducting experiments with a sample size of n = 3 to examine statistically significant differences among multiple subject groups, as we did in our study [4, 7]. The second issue is related to the technical difficulty of our procedure and ethical concerns for experimental animals. In a mouse cochlea, the volume of perilymph is extremely small, approximately 5 μL [8]. Therefore, collection of this fluid with minimal contamination of other fluids and cells is technically difficult and requires a high level of skill. Indeed, our preliminary experiments carried out before this study with commercially available wild-type mice showed that the success rate of the collection was limited to ~20%. In this context, we need to pay attention to such ethical concerns that, in general, the number of animals used in any study must be minimized as much as possible. 

Based on the two issues mentioned above, we concluded that the sample size described in our work is acceptable. Nevertheless, in prospect, significant improvement of the skill for sample collection will improve the success rate for the perilymph collection, increase the sample size, and enable larger-scale analysis. We hope that such technical advances will be realized by future endeavors to address the limitations of this study. In the revised version of our manuscript, we discussed this issue on page 16 line 344 – page 17 line 356 in the Discussion section.

4. Can you discuss any potential confounding factors that may have influenced the observed protein changes in the cochlear perilymph?

 Thank you for your constructive comment. We were careful to avoid including confounding factors as much as possible. Although the samples collected from mice of both sexes were mixed and analyzed (Fig. 4), in the animals the genetic background (C57BL/6J) and ages (3, 6, and 12 months old) were the same. Therefore, we do not think this issue was a significant confounding factor in our experimental approach. Another factor could be attributed to the technical difficulty of the experimental procedure. Because the volume of perilymph in a cochlea is small, in the sampling of this fluid we cannot completely rule out the possibility of contamination of cellular components and/or other extracellular solutions. We additionally described these possibilities in the revised manuscript (page 16, lines 337 – 344) in the Discussion section.

5. Can you provide insights into the mechanisms underlying the observed alterations in protein composition in the cochlear perilymph in response to AD-like pathology?

 We appreciate your question. AD is a systemic disease and affects the profiles of not only cerebrospinal fluid but also circulating blood [1, 2]. In the cochlea, perilymph is exposed to numerous capillaries [9] as well as being connected to cerebrospinal fluid directly and physically through the cochlear aqueduct, a narrow tubule [10-12]. Therefore, it is possible that the perilymph in the AD mouse model is influenced by factors in modified blood plasma and/or alternatively those in cerebrospinal fluid. Considering the latter route, when the perilymphatic protein profile that we obtained in AppNL-G-F/NL-G-F mice was compared to the data of the CSF from the same mouse model in the literature [1], only a few proteins were overlapped in these two subjects (see page 13, lines 266 – 269 and page 15, lines 325 – 326). Therefore, the exchange of proteins between the perilymph and cerebrospinal fluid is unlikely to be active. Nevertheless, the cochlear aqueduct may allow the passage of small molecules of a molecular weight of <5 kDa [13]. In this context, some metabolites up-regulated in the brain by Aβ-pathology may be transmitted to the perilymph, affecting cochlear cells, and modulating the protein profile of the fluid.

 These possibilities were additionally described in the Discussion section of the revised version (page 16, lines 333 – 334).

6. How do you envision translating these findings into potential therapeutic strategies or diagnostic biomarkers for AD-related hearing impairment?

 Yes, this is a crucial issue. As we mentioned in our response to Question 2, the change of perilymphatic protein landscape in the AD mouse model may increase vulnerability to different types of hearing loss. In this context, a possible strategy for clinical implementation may be the application of our findings to the establishment of a diagnostic biomarker for AD-related hearing impairment. In other words, the collection of cochlear perilymph from AD patients at an early stage and analysis of the protein profile of this fluid may partially predict the development or acceleration of hearing loss. Of note, under the current clinical situation, implementing this strategy is challenging. In the process of sampling the perilymph from the cochlea in humans, the eardrum should be, at least in part, detached and a fine needle must be inserted into the cochlea through the round window membrane. These existing invasive techniques should be improved to reduce the damage to the ear. Along this line, the collection volume of the perilymph must be reduced to minimize additional hearing impairment. For such purposes, the sensitivity of MS analysis should be improved. If all the issues mentioned above are realized, then analysis of the protein landscape in perilymph might serve as a diagnostic biomarker for AD-related hearing impairment in clinical practice. We added a description of this point to the Discussion section in the revised manuscript on page 17 lines 364 – 368.

7. Have you considered conducting additional experiments to validate the observed protein changes using alternative methods or approaches?

We thank you for your valuable comment; we agree with your concern. We think that western blot and/or ELISA analyses will validate our findings. Note that, each of these approaches will require a large volume of perilymph sample. As mentioned above (see our response to Question 3), in this study we made every effort to minimize the number of mice used for our experiments due to technical difficulties and ethical concerns. Therefore, we were unable to carry out western blot analysis or ELISA assays.

 In the revised version, this issue was described as one of the limitations of our study (see Discussion section; page 17, lines 356 – 359).

8. Could you discuss the relevance of the observed protein changes in the cochlear perilymph to other neurodegenerative diseases beyond AD?

 We appreciate your constructive comment. A few neurodegenerative diseases are likely associated with hearing loss [14, 15]. For example, asymptomatic auditory dysfunction is a non-motor manifestation of early-onset Parkinson’s disease [14]. Given that this disease is related to �-synuclein deposits in the brain, the perilymph protein profile may change. Indeed, in this study � -synuclein was found in perilymph although little change was detectable between the samples of wild-type and AppNL-G-F/NL-G-F mice. In the revised text, we added this discussion to page 17 lines 370 – 374.

9. What are the next steps in your research agenda to further elucidate the relationship between AD pathology and cochlear dysfunction, based on the findings of this study?

Thank you for your question. For the next steps, we would like to identify the mechanisms underlying the up- and down-regulation of the perilymphatic proteins and the pathophysiological roles of these changes on cochlear activity and even brain function. In particular, the latter issue must be crucial from a clinical standpoint as well. In this context, we could generate conditional transgenic mice that overexpress or lack the target proteins in the cochlea in our future work. This prospect has been described in the revised manuscript (page 17, lines 361 – 364).

Finally, in the revised version, we have modified some sentences and words to improve the text.

Response to Reviewer #1

Reviewer #1: The author in this research study is trying to establish correlation between Alzheimer’s disease and hearing impairment by inducing disease in the animal model, The experimental work conducted is satisfactory. However further studies can be conducted to human beings to substantiate the finding s and confirm the proof of concept.

We appreciate your positive and constructive comments very much. We agree with your suggestion that further studies w

---

## [Editor Report · Decision Letter 1]

24 Apr 2024

Disturbance in the protein landscape of cochlear perilymph in an Alzheimer's disease mouse model

PONE-D-24-02136R1

Dear Dr. Hiroshi Hibino,

We’re pleased to inform you that your manuscript has been judged scientifically suitable for publication and will be formally accepted for publication once it meets all outstanding technical requirements.

Kind regards,

Abdelwahab Omri, Pharm B, Ph.D, Laurentian University

Academic Editor

PLOS ONE

---

## [Editor Report · Acceptance letter]

29 Apr 2024

PONE-D-24-02136R1 

PLOS ONE

Dear Dr. Hibino, 

I'm pleased to inform you that your manuscript has been deemed suitable for publication in PLOS ONE. Congratulations! Your manuscript is now being handed over to our production team.

Kind regards, 

on behalf of

Dr. Abdelwahab Omri 

Academic Editor

PLOS ONE